# Photoinduced dynamics during electronic transfer from narrow to wide bandgap layers in one-dimensional heterostructured materials

Yuri Saida[1], Thomas Gauthier[2], Hiroo Suzuki [3] ✉, Satoshi Ohmura [4] ✉, Ryo Shikata[1], Yui Iwasaki[1], Godai Noyama[1], Misaki Kishibuchi[3], Yuichiro Tanaka[3], Wataru Yajima[1], Nicolas Godin[2], Gaël Privault[2], Tomoharu Tokunaga[5], Shota Ono[6], Shin-ya Koshihara[7], Kenji Tsuruta [3], Yasuhiko Hayashi [3], Roman Bertoni [2] ✉ & Masaki Hada [8] ✉

Electron transfer is a fundamental energy conversion process widely present in synthetic, industrial, and natural systems. Understanding the electron transfer process is important to exploit the uniqueness of the low-dimensional van der Waals (vdW) heterostructures because interlayer electron transfer produces the function of this class of material. Here, we show the occurrence of an electron transfer process in one-dimensional layer-stacking of carbon nanotubes (CNTs) and boron nitride nanotubes (BNNTs). This observation makes use of femtosecond broadband optical spectroscopy, ultrafast time-resolved electron diffraction, and first-principles theoretical calculations. These results reveal that near-ultraviolet photoexcitation induces an electron transfer from the conduction bands of CNT to BNNT layers via electronic decay channels. This physical process subsequently generates radial phonons in the one-dimensional vdW heterostructure material. The gathered insights unveil the fundamentals physics of interfacial interactions in low dimensional vdW heterostructures and their photoinduced dynamics, pushing their limits for photoactive multifunctional applications.

Many photofunctional materials and molecules in synthetic, industrial, and natural systems rely on electron or charge transfer processes. For example, electron transfer at chlorophylls in biological systems is important in converting energy from photo-to-chemical energy during photosynthesis[1]. Another famous case is the electron transfer between the retinal chromophore in rhodopsin, which is fundamental for eye vision[2–4]. In the industrial field, the pn-junction in solar cells, where the photoinduced electron transfer occurs, is the fundamental building block of semiconductor devices. The reverse process is also used for light-emitting diodes[5]. In synthesis, photocatalytic reactions initiated by the absorption of photoenergy are demonstrated by the subsequent chemical reactions[6,7]. Actually, electron transfer processes in

[1]Graduate School of Science and Technology, University of Tsukuba, Tsukuba 305-8573, Japan. [2]Univ Rennes, CNRS, IPR (Institut de Physique de Rennes)—UMR 6251, F-35000 Rennes, France. [3]Graduate School of Environmental, Life, Natural Science and Technology, Okayama University, Okayama 700-8530, Japan. [4]Faculty of Engineering, Hiroshima Institute of Technology, Hiroshima 731-5193, Japan. [5]Graduate School of Engineering, Nagoya University, Nagoya 464-8603, Japan. [6]Institute for Materials Research, Tohoku University, Sendai 980-8577, Japan. [7]School of Science, Tokyo Institute of Technology, Tokyo 152-8551, Japan. [8]Institute of Pure and Applied Science and Tsukuba Research Center for Energy Materials Science (TREMS), University of Tsukuba, Tsukuba 305-8573, Japan. ✉e-mail: hiroo.suzuki@okayama-u.ac.jp; s.ohmura.m4@cc.it-hiroshima.ac.jp; roman.bertoni@univ-rennes.fr; hada.masaki.fm@u.tsukuba.ac.jp

van der Waals (vdW) heterostructures are a very attractive topic due to the unique characteristics originating from stacking low dimensional materials[8–13]. Apart from these new static functions of heterostructures, it is also interesting to investigate the functions of stacked low-dimensional materials in the photoexcited non-equilibrium state. To understand photoinduced electronic dynamics in vdW heterostructures, ultrafast transient spectroscopy, ultrafast transient microscopy, and time- and angle-resolved photoemission spectroscopy have been applied for two-dimensional (2D) vdW heterostructures[14–23]. Ultrafast time-resolved electron diffraction (UED) measurements and first-principles calculations have been used to investigate electron-phonon coupling in a WSe$_2$/WS$_2$ vdW heterostructure, where electron transfer between WSe$_2$ and WS$_2$ drives phonon emission in both layers at femtosecond timescale[24]. One-dimensional (1D) van der Waals (vdW) heterostructures[25–27] emerge as a new class of materials in which different atomic layers are coaxially stacked. The anisotropic energy transfer between layers of 1D vdW heterostructures is being applied for various electronic devices[28–32], differing from those of 2D vdW heterostructures. Actually, it is challenging to monitor concomitantly interlayer electronic and phononic dynamics of 2D materials using UED. Indeed, the very high kinetic energy of electrons translates into an almost flat Ewald sphere, thus inducing a much lower sensitivity of out-of-plane/interlayer dynamics.

Here, one-dimensional systems provide an easier and more direct access to interlayer dynamics via UED measurements. In this study, we demonstrate the ultrafast photoinduced charge transfer through electronic channels from carbon nanotubes (CNTs) to boron nitride nanotubes (BNNTs), subsequently generating radial phonons at the interface of the 1D vdW heterostructure. These observations are allowed by the complementary use of femtosecond optical spectroscopy and UED. Overall, our results demonstrate the existence of unusual electron transfer channels generated by the layer-stacking of CNTs and BNNTs, which is also supported by first-principles theoretical calculations.

## Results and discussion
### One-dimensional vdW structure and experimental design
Figure 1 depicts the global concept of this study where the photoinduced dynamics of 1D CNT-BNNT vdW heterostructure is investigated by femtosecond absorption spectroscopy and UED. Few-walled CNTs[33] are covered with few- and multi-walled BNNTs[34] in a 1D vdW heterostructure fashion (Supplementary Fig. 1). Thorough characterization of the samples including transmission electron microscopy (TEM), electron energy loss spectroscopy (EELS), and Raman spectroscopy are presented in Supplementary materials (Supplementary Figs. 2–9). According to these fundamental characterizations, the estimated 1D-vdW heterostructure consists of CNTs with a diameter of ~5 nm (outer diameter) and BNNTs with a diameter of ~6 nm (inner diameter). CNTs with semiconducting and metallic characters absorb light in the ultraviolet (UV), visible, and infrared (IR) spectral ranges. BNNTs with a large optical bandgap of approximately 6 eV present electronic excitation in the deep-UV light (Supplementary Fig. 10). According to density functional calculations of a three-walled CNT (Supplementary Fig. 11 and Supplementary Table 1), the electronic band structure of the few-walled CNTs differs from those of a purely 1D with van Hove peaks but is similar to those of semiconductors. For both experiments, only the CNT subpart of material is photoexcited by near-UV light (3.1 eV, 400 nm). Upon near-UV photoexcitation, broadband transient absorption spectroscopy spanning from deep-UV to near-IR spectral range (0.7–5.4 eV) is sensitive to electron transfer

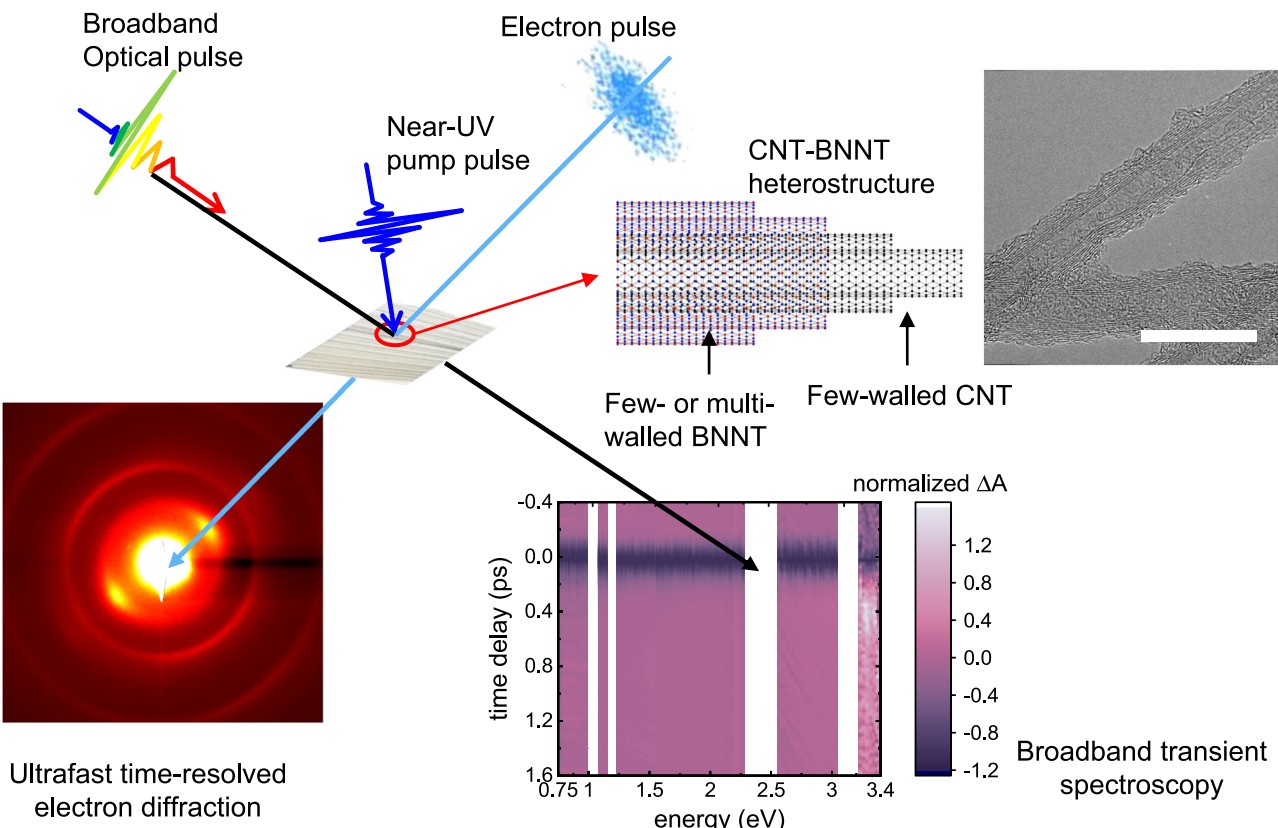

**Fig. 1 | Dynamics induced by phototriggered electron charge transfer probed with different time-resolved techniques.** The heterostructure is composed of few-walled carbon nanotubes (CNTs) in the inner shell and few- and multi-walled boron nitride nanotubes (BNNTs) in the outer shell. Electron and phonon dynamics in CNT-BNNT heterostructure are monitored by ultrafast broadband transient absorption spectroscopy and ultrafast time-resolved electron diffraction (UED). The transmission electron micrograph of the CNT-BNNT 1D vdW heterostructure is in the inset. The white scale bar in the micrograph corresponds to 20 nm.

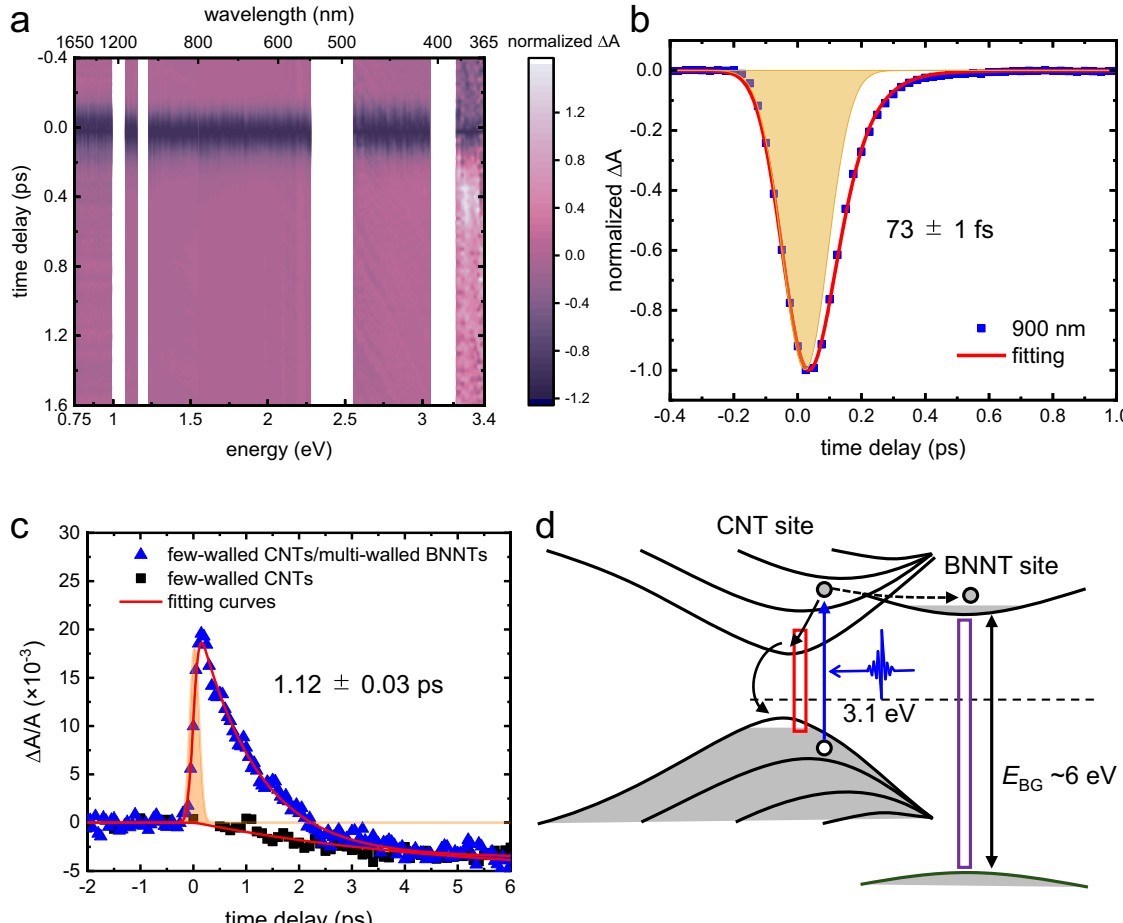

**Fig. 2 | Broadband transient absorption spectroscopy of CNT-BNNT hetero-structure following femtosecond photoexcitation. a** Normalized transient absorption spectra of CNT-BNNT heterostructure in the visible-to-IR region. **b** Single wavelength time trace at 900 nm. The yellow shaded area represents the instrumental response function (IRF). The short-life component in this wavelength is approximately 73 fs. **c** Transient absorption time traces of CNT-BNNT heterostructure and CNTs probed at 230 nm (5.4 eV). The yellow shaded area represents

the IRF. The transient absorption spectra in **b** and **c** are fitted by Eq. (1). **d** Schematic illustration of near-UV photoinduced electronic dynamics in CNT-BNNT heterostructures. $E_{BG}$ is the bandgap energy. The blue arrow indicates the photoexcitation of CNTs. The black solid and broken arrows show the electronic relaxation in the CNT and electron transfer from the CNT to BNNT, respectively. The red and purple rectangles show probing photoenergies of approximately 900 nm and 230 nm, respectively.

from CNTs to BNNTs and the electronic relaxation dynamics within CNTs. In parallel, time-resolved electron diffraction measurements[35] monitor how the electron transfer process from CNTs to BNNTs induces structural dynamics and, more precisely, the emission of radial phonons in the material.

**Ultrafast transient photoresponse**

The broadband transient absorption spectrum of CNT-BNNT hetero-structure under near-UV photoexcitation (400 nm, 3 mJ cm$^{-2}$) with probe spectral range of 360–1600 nm is shown in Fig. 2a. BNNTs do not present electronic resonance around 400 nm; therefore, the photo-driven nonthermal electronic dynamics is initially set inside the CNTs. Over the visible and IR spectral range (0.75 to 3.35 eV), the transient absorption ($\Delta A$) is negative right after photoexcitation. This feature might be due to the generation of free carriers and excitons inside CNTs and can be understood as a transient manifestation of Pauli blocking thus reducing the probability of electronic transitions. In addition, the linear character of the photoresponse (Supplementary Fig. 12) and its temporal asymmetry (Fig. 2b) exclude multi-photon absorption from contributing to the observed phenomenon. After, the optical absorption relaxes back to its initial value within 1 ps (Fig. 2b), and its time-dependent dynamics are fitted with the following

equation as a function of time ($t$):

$$
\begin{aligned}
\Delta A = A_1 \cdot &\frac{\left[\text{erf}\left((t/T - T/\tau_1)/\sqrt{2}\right) + 1\right]}{2} \cdot \exp\left[\frac{(T/\tau_1)^2}{2} - \frac{t}{\tau_1}\right] \\
+ A_2 \cdot &\frac{\left[\text{erf}\left((t/T - T/\tau_2)/\sqrt{2}\right) + 1\right]}{2} \cdot \exp\left[\frac{(T/\tau_2)^2}{2} - \frac{t}{\tau_2}\right],
\end{aligned}
\tag{1}
$$

with the following parameters: a long-life component $A_1$, the amplitude of change in transient absorption (a short-life component) $A_2$, experimental time-resolution $T$, and the time constant of recovery time for the long- and short-life components $\tau_1$ and $\tau_2$. The initial electronic relaxation process occurs in the electronic band structures of CNTs; therefore, the transient spectra of the CNT-BNNT hetero-structure in the visible-infrared ranges are identical to that of pure CNTs (Supplementary Fig. 13). According to the steady state optical properties of CNT-BNNT (Supplementary Fig. 10) probing with deep-UV light at 230 nm (5.4 eV) should provide insights about electronic dynamics occurring between CNTs and BNNTs layers (Fig. 2c). Indeed, the optical absorption promptly increases within ~0.5 ps and relaxed back initial value in ~1.1 ps according to the fit using Eq. (1). This feature

is not observed in pure CNTs (Fig. 2c), suggesting that the observed dynamics can be attributed to the charge transfer process inside the 1D vdW heterostructure.

Regarding its physical origin, this transient increase in absorption may be caused by the band renormalization effect occurring in BNNTs as previously reported for other vdW materials[36]. Actually, in other vdW heterostructures such as graphene/MoS$_2$ and graphene/WS$_2$, the transient absorption around the optical bandgap decreases, while

photon energies below or above the bandgap present an increase of absorption via the band renormalization effect induced by electron transfer[16,37]. An intertube excitation across the CNT and BNNT interface is another possible mechanism to describe the physical phenomena into play[38]. However, our experimental observation could not discriminate between these two phenomena since the results are similar after the dissociation of the intertube exciton. Figure 2a also shows a small positive residual signal after the negative peaks in the UV region

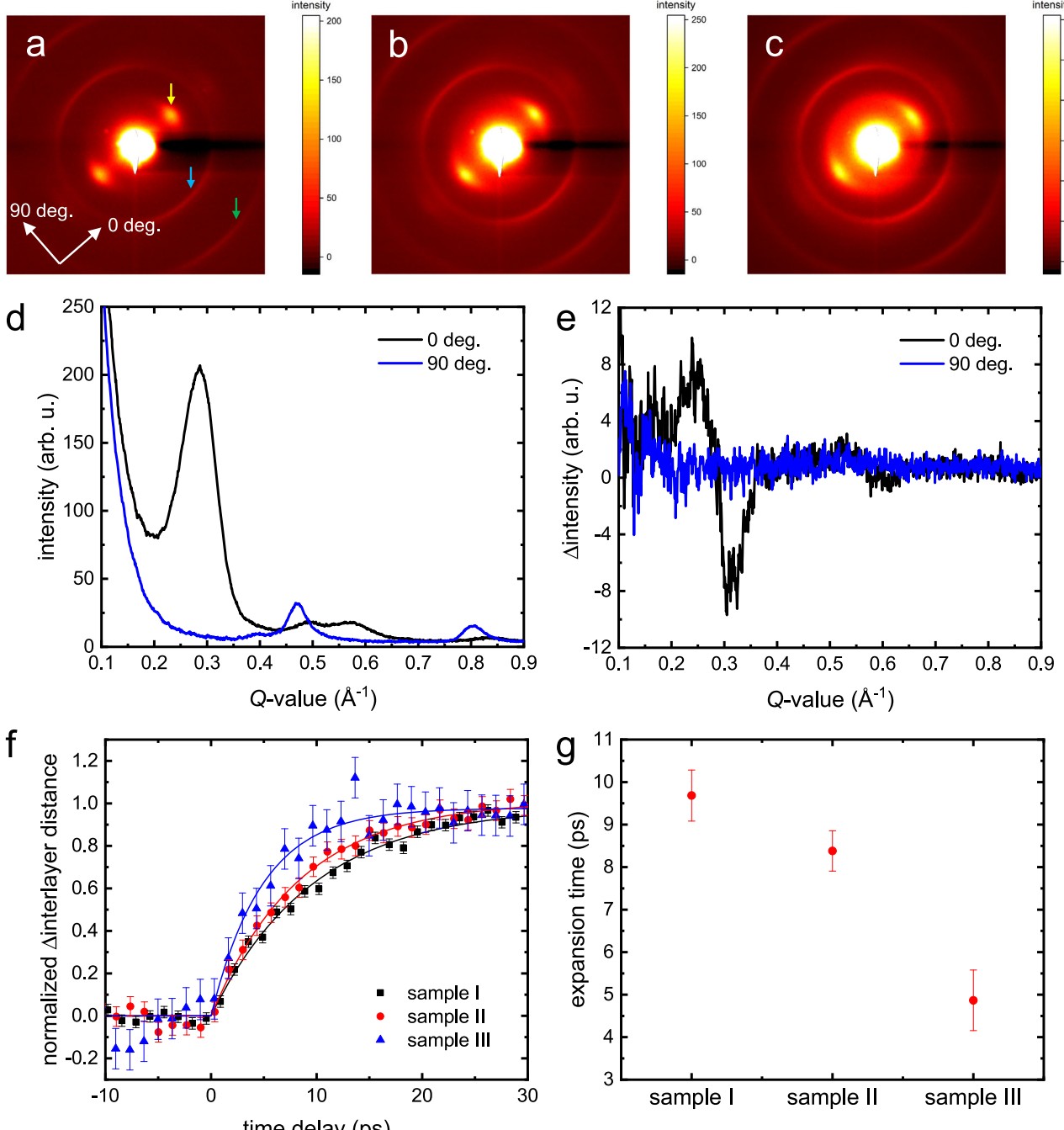

**Fig. 3 | Ultrafast time-resolved electron diffraction measurements on few-walled CNTs and CNT-BNNT heterostructures following 400 nm excitation.** Electron diffraction patterns from samples I (**a**), II (**b**), and III (**c**) before photo-excitation. **d** Radial average of the diffraction pattern from sample I. In the legend, 0 deg. is the direction to the broad spots as indicated in the inset of (**a**), and 90 deg. is parallel to the long axis of CNTs. The yellow arrow shows the diffraction spots from the interlayer ordering. The blue and green arrows show diffraction rings

from the (100) and (110) planes of CNTs or BNNTs. **e** Differential diffraction pattern from sample I at the time delay of +150 ps. **f** Time evolution of normalized inter-layer distance observed in **a**–**c**. Solid lines are fitting curves obtained with Eq. (2). The error bars represent the standard deviation at each time delay obtained from the negative delay time points ($n = 12$). **g** Time constants of interlayer expansion induced by photoexcitation. The error bars represent the standard deviation derived from the lines of best fit ($n = 41$), as shown in **f**.

above the energy of the pump pulse (~365 nm). This feature is similar to the transient absorption dynamics at the wavelength of 230 nm, which shows a positive peak and subsequent negative residual signal. Thus, the band renormalization or intertube exciton may be responsible for this signal in UV region (~365 nm). The near-UV pump light induces valence to conduction band excitation in CNTs. CNTs are intrinsic p-type semiconductors[39]. Because we can inject electrons into the electronic bands of a BNNT layer with 400 nm light pump (3.1 eV) and excluding a two-photon absorption process, there should be a local contact between the electronic bands of CNT and BNNT. As consequence, the electronic transfer occurs through the interface between the CNTs and BNNTs and pn-junctions are locally created at their interface. The interlayer electronic transfer is observed when probing in the deep-UV (230 nm, 5.4 eV) with sensitivity to both CNTs and BNNTs electronic structures. Figure 2d summarizes in a simplified way the near-UV photoinduced electronic dynamics of CNT-BNNT heterostructure from the viewpoint of the electronic band structure. A part of the photoexcited electrons transfer to the conduction band of BNNTs through an electron transfer channel provided by the low-dimensional character of the vdW heterostructures. Without holes in the valence band of BNNTs, electron–hole recombination might not occur inside BNNTs. The transferred hot electrons can release their excess energy by generating phonons (electron-phonon coupling) within BNNTs layers.

## Ultrafast structural dynamics

Time-resolved diffraction experiments can directly access the photoinduced structural rearrangements of materials[40–42]; therefore, this methodology is advantageous to observe phonon dynamics triggered by electron transfer in the CNT-BNNT heterostructure. Due to the very flat Ewald sphere of the electron at a kinetic energy of 75 keV, measurements in transmission geometry are mainly sensitive to in-plane motions. This feature precludes a thorough investigation of interlayer atomic motions in 2D materials while operating in normal geometry. On the contrary, this limitation is lifted when studying well-aligned 1D materials, allowing the monitoring of atomic interlayer dynamics. This is the case for the well aligned 1D vdW heterostructures under study.

Electron diffraction patterns from CNTs and CNT-BNNT heterostructures at equilibrium state are shown in Fig. 3a–c, respectively. We label the few-walled CNTs and few-walled CNTs covered with few- and multi-walled BNNTs samples as samples I, II, and III, respectively. As shown in the patterns, clear diffraction rings and broad peaks were observed. In Fig. 3a, the broad peaks correspond to the interlayer of CNTs. The radial averages of the diffraction pattern from sample I are shown in Fig. 3d. The first and second-order rings are diffracted from the (100) and (110) planes, respectively, which suggests that the few-walled CNTs sample contains all types of CNT-chirality as described by a previous report[43]. Diffraction features related to interlayer distances and bond lengths appear at almost the same positions in reciprocal space due to the structural similarity between CNTs and BNNTs. Consequently, similar diffraction patterns are observed in the 3 sample types, as shown in Fig. 3a–c. In Fig. 3b, c, a diffuse scattering ring appears from the atomic layer of BNNTs and is caused by the random growth of BNNTs. However, the broad Bragg peaks observed in the figures originate from the coaxially oriented atomic layers of CNTs, CNT-BNNT, and BNNTs. Therefore, we use these diffracted features, Debye–Scherrer rings and broad Bragg peaks, to analyse the photoinduced structural dynamics.

The scattering vector value (Q) of the broad Bragg peaks directly relates to the average lattice distance in dual space. Therefore, one can calculate the photoinduced changes in the average interlayer distance from the time evolution of the position in reciprocal space. The experimental conditions for optical pumping of both UED and optical measurements are similar, i.e., wavelength of 400 nm, polarization axis

parallel to the long axis of CNTs, and excitation density of 3 mJ cm$^{-2}$. The differential diffraction pattern (after–before the photoexcitation) of sample I is shown in Fig. 3e, where the peak shift toward a lower Q-value (expansion of interlayer distance) is observed. Figure 3f shows the time evolution of normalized interlayer distance (Δd) of few-walled CNTs and CNT-BNNT heterostructures following ultrafast photoexcitation. Solid lines in Fig. 3f indicate fitting curves with the following equation as a function of time (t):

$$\begin{cases} \Delta d = 0 & (t < 0) \\ \Delta d = B\left[1 - \exp\left(-\frac{t}{\tau}\right)\right] & (t \geq 0) \end{cases}, \tag{2}$$

where B and τ are the amplitude and time constant of changes in interlayer distance, respectively. A previous study already described the mechanism behind the photoinduced radial phonon dynamics in few-walled CNTs[35]. Upon photoirradiation, excited electrons relax through electron-phonon coupling in CNTs and emit longitudinal, transverse, distorted, and radial phonon modes. Transverse and radial phonon modes induce CNTs vibrations in the radial direction. This global dynamic of incoherent nature results in an increase in the average interlayer distance. Similar ultrafast structural dynamics also occur in samples II and III. The radial averages and differential diffraction patterns for samples II and III are shown in Supplementary Fig. 14. One should note that photoexcitation induces vibrations within all the layers in CNTs and BNNTs for samples I, II, and III (Supplementary Fig. 15). Figure 3g shows the time constants for the interlayer expansion extracted from Fig. 3f. The expansion time of samples I, II, and III are 9.7 ± 0.6 ps, 8.4 ± 0.5 ps, and 4.9 ± 0.7 ps, respectively. The interlayer expansion time is faster in the materials with BNNT layers. It is worth mentioning that only the CNT layers absorb the near-UV light. Considering a case where the initial absorbed photon energy at the CNTs side is transferred to BNNTs only in the form of simple heat, one could expect to have in the fastest scenario, a joined expansion of both BNNT and CNT mimicking the dynamics of bare CNTs. Thus, the observed phenomenon cannot be explained by simple heating effects in the system and pushes the need to consider the fast electron transfer as one of the key players for the observed ultrafast structural dynamics. This counterintuitive phenomenon can be explained by considering the electron transfer process discussed previously. We would like to stress that the amplitude of the photoinduced dynamics in samples I, II, and III have a linear response with respect to the excitation density (Supplementary Fig. 16). Also, BNNTs do not absorb the 400 nm light pulses (Supplementary Fig. 17) and therefore are not directly photoexcited.

By considering the transient optical absorption results, electron transfer from CNTs to BNNTs in the 1D CNT-BNNT heterostructure occurs within 0.5 ps, and the relaxation process occurs in ~1 ps. While there are free charges in the conduction band of the BNNT, these hot electrons can release their energy by emitting phonons within the BNNT side within 1 ps. This 1 ps timescale refers to the dynamics of the abrupt transient optical change monitors at 230 nm. This effect might accelerate the emission of radial phonons in the 1D CNT-BNNT heterostructures compared to bare CNTs. The charge delocalization between CNTs and BNNTs induced by the electron transfer would also affect the radial phonon generation, which can be observed in plane motion from the changes of C–C and B–N average bond lengths (see Supplementary Fig. 18).

## First principles theoretical calculations

For further exploration of the electron and phonon dynamics of CNT-BNNT heterostructure under photoexcitation, theoretical calculations based on time-dependent density functional theory (TD-DFT) are performed on a model structure of graphene and hexagonal boron nitride (graphene/h-BN) and CNT and BNNT (CNT/BNNT) heterostructures (see the details in method section and Supplementary

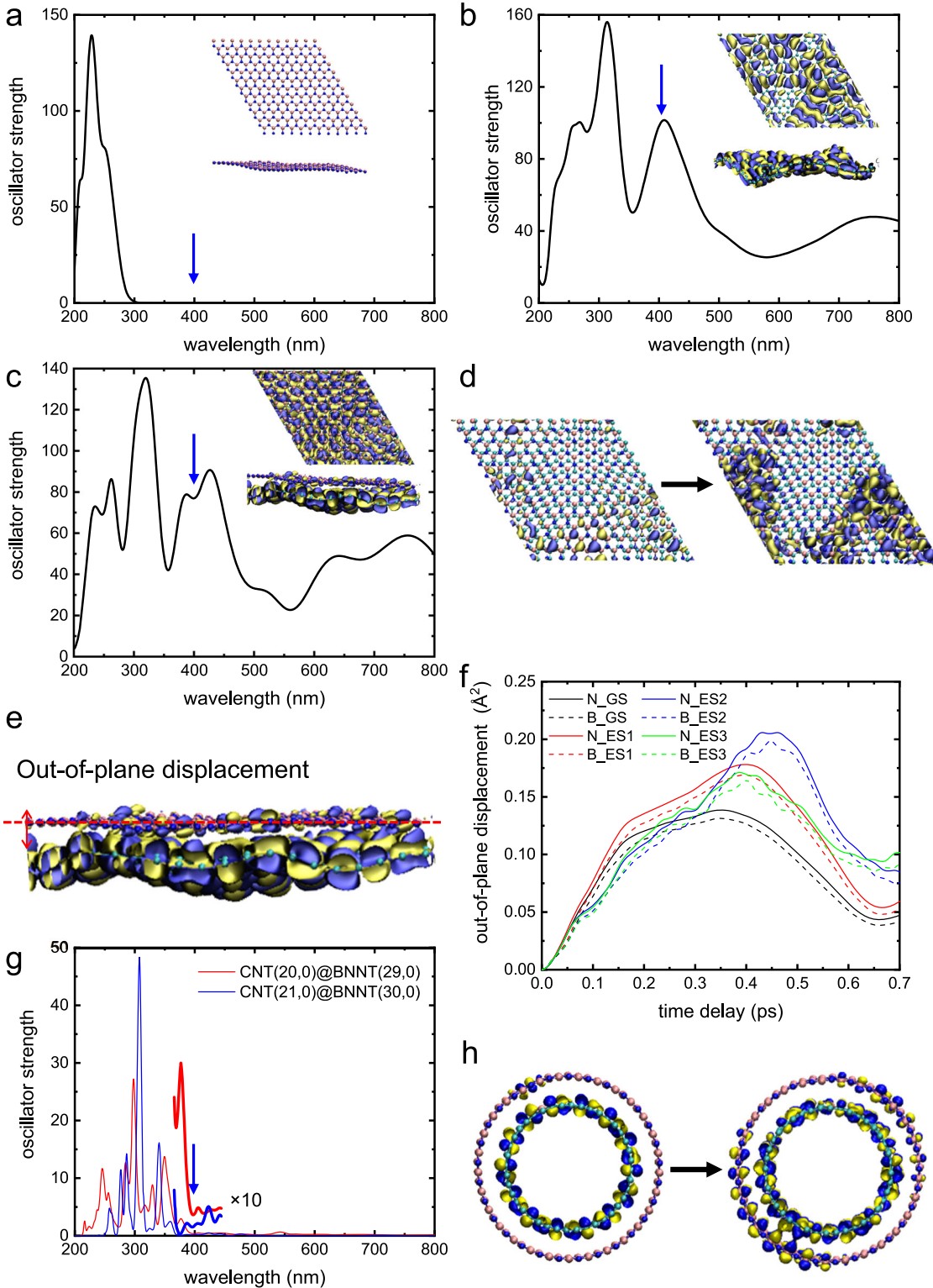

**Fig. 4 | First-principles calculations of photoinduced dynamics in graphene/h-BN and CNT/BNNT heterostructures.** Oscillator strength for the electronic transition for h-BN (**a**), graphene (**b**), and graphene/h-BN heterostructure (**c**) calculated based on TDDFT. Inset figures show the wavefunctions for electronic transition at an energy of 3.1 eV. The wavefunctions represent the Kohn–Sham orbitals at the unoccupied states corresponding to the excited states in the transition at the peaks of the oscillator strength, where the yellow and blue colors represent the isosurfaces of the wavefunctions with the values of 0.002 and −0.002 atomic unit, respectively. **d** Electronic wavefunction of the h-BN layer at an energy of 3.1 eV of graphene/h-BN heterostructure before and after photoexcitation of graphene layer with an energy of 3.1 eV. For the sake of simplicity, the wavefunction around h-BN is displayed. **e** Schematic illustration of out-of-plane displacement in h-BN layer. **f** Mean square values of out-of-plane displacement of boron (B) and nitrogen (N) atoms before and after photoexcitation. GS and ES are in the ground state and excited state. The calculation for the excited state was performed three times and the ES1, ES2, and ES3 in the caption indicate the results of the first, second, and third calculations. **g** Oscillator strength for the electronic transition for CNT/BNNT heterostructures with the interfacial CNT diameter of approximately 1.6 nm. The pair of integers (*n*, *m*) in the figure caption are chiral index. **h** Electronic wavefunction of a CNT/BNNT heterostructure at an energy of 3.1 eV before and after photoexcitation.

Figs. 19–21). Since the diameter of the CNTs used in our experiment is large (~5 nm) compared with single-walled CNTs, the 2D approximation translates as a tangent plane of the cylindrical 1D vdW heterostructure. First of all, Fig. 4a–c shows the oscillator strength for the electronic transition for h-BN, graphene, and graphene/h-BN heterostructure calculated in the ground state as a basis of plane wave. The electronic transition at an energy of 3.1 eV (a wavelength of 400 nm) is not present on h-BN but observed on graphene.

Figure 4c shows oscillator strength corresponding to the absorption calculated from graphene/h-BN heterostructure. An absorption peak of sitting around 300 nm originates from the optical band gap of h-BN. The other peaks at lower energy originate from the graphene layer. As shown in Fig. 4c, a slight penetration of wavefunction is induced on the h-BN layer in the graphene/h-BN heterostructure for an energy of 3.1 eV in the ground state, which should correspond to the electron charge transfer channels discussed previously. A significant spatial modulation of the electronic wavefunction of graphene and h-BN layers occurs upon photoexcitation of the graphene layer with a near-UV light (Fig. 4d). The wavefunction represents the Kohn−Sham orbitals at the unoccupied states corresponding to the excited states in the transition at the peaks of the oscillator strength, where the yellow and blue colors represent the isosurfaces of the wavefunctions with the values of 0.002 and −0.002 atomic unit, respectively. The simulation presents an overlap of wavefunctions of graphene and h-BN from the ground state to the excited state via the excitation of graphene layer using 400 nm pulses. This result suggests the presence of an electron transfer from the initially photoexcited graphene layer to the unperturbed h-BN layer. To understand the phonon dynamics of the graphene/h-BN heterostructure, we defined the displacement of boron and nitrogen atoms from the h-BN layer in the interlayer direction as an out-of-plane displacement (Fig. 4e). We calculated the mean square values of the out-of-plane displacement in the ground state and the excited state as shown in Fig. 4f, when excited state is defined as the case where one electron in the graphene layer is excited. The mean out-of-plane displacement in the excited state is larger than that in the ground state after 0.2–0.3 ps, reflecting that the lattice of h-BN is more displaced along with the photoexcitation of the graphene layer. This displacement in the h-BN layer occurs quite fast within <1 ps after the initial photoexcitation of the graphene layer, which is slightly faster than the experimentally observed results.

However, the underlying dynamics imply that the electron transfer from the graphene layer to the h-BN layer plays an important role in the phonon dynamics agree well with the results from UED measurements.

The first-principles calculations were also performed on 1D vdW interface between a CNT and a BNNT with various diameters (Supplementary Figs. 20 and 21). The absorption band at a photon energy of 3.1 eV (400 nm) does not appear in the interface between small diameter (less than 1 nm); therefore, the electrons cannot transfer from CNT to BNNT with small diameter at the interface (Supplementary Fig. 22). However, above the diameter of 1.6 nm, absorption shifts to a wavelength close to 400 nm at the 1D vdW interface (Fig. 4g). The chiral index[44] (n, m) of the larger diameter 1D-vdW heterostructures are CNT(20, 0)@BNNT(29, 0) and CNT(21, 0)@BNNT(30,0). As shown in Fig. 4h, a similar effect as the one observed in a 2D interface between graphene and boron nitride is produced at CNT and BNNT interface, with occurrence of the interlayer electron transfer. To understand the stacking effects of CNTs on the electron transfer between a CNT layer and a BNNT layer, we also performed the first-principles calculations on a double-walled CNT covered with a single-walled BNNT (Supplementary Fig. 21), which shows similar electron transfer effects to a single-walled CNT covered with a single-walled BNNT (Supplementary Figs. 23 and 24).

## Physical picture of ultrafast dynamics in vdW heterostructure

We rationalized the physical picture obtained from our experimental results coupled with theoretical calculations in Fig. 5, where the ultrafast out-of-equilibrium electron and phonon dynamics are unveiled when considering the electron transfer channel between the interface of 1D vdW heterostructures. Transient optical spectroscopy directly monitors the electron transfer from CNTs to BNNTs occurring in ~0.5 ps through an electronic channel created between the heterostructure layers. In parallel, time-resolved electron diffraction reveals a faster phonon emission in the BNNTs subpart triggered by the electronic relaxation. This statement is corroborated by first-principles calculations based on TDDFT that support the creation of unusual electron transfer channels between graphene and boron nitride layers. In this case, the electron transfer directly induces a fastest increase in the interlayer distance compared to bare CNT, resulting in a new phonon distribution in the BNNT layers. This electron transfer may be induced by different mechanisms including the generation of

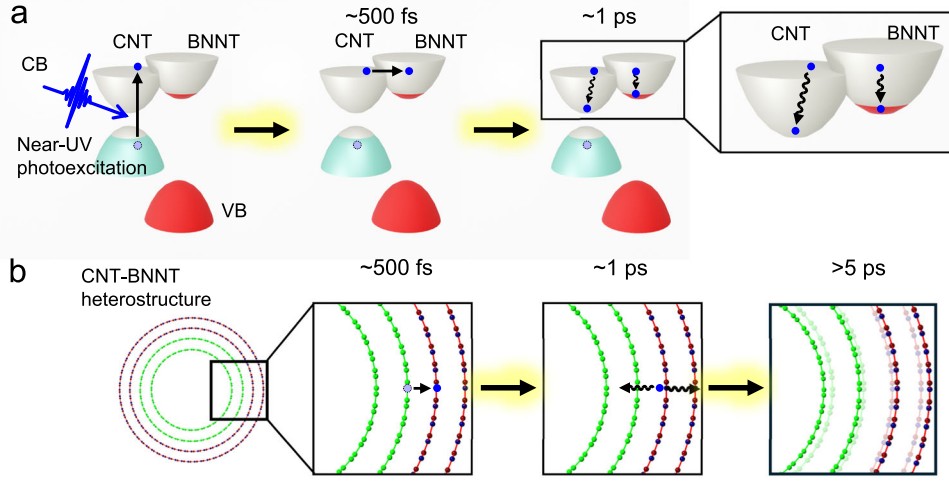

**Fig. 5 | Electron relaxation in the CNT and BNNT layers and phonon emission in the CNT-BNNT 1D vdW heterostructure. a** Near-UV light is absorbed by the CNT subpart, and hot-electron generated at CNT subpart is transferred to BNNT subpart in approximately 500 fs. The hot electrons relax to the bottom of the conduction band emitting phonons within the BNNT side within 1 ps. **b** The generated phonon expands the average interlayer distance of CNTs and BNNTs in approximately 5 ps. In this picture, we only show the pathway to electron transfer from CNTs to BNNTs and phonon generation within BNNTs. Other electronic or phononic dynamics may occur with different dissipation channels. VB and CB are the valence band and conduction band, respectively.

intertube exciton[38] or π-plasmonic interaction[45] at the interface between CNT and BNNT layers, which induces Coulomb interaction between the heterolayers. According to the theoretical calculations, the local vibrational motions occur in <500 fs, where some parts of the distance between the CNTs and BNNTs decrease, and the others increase. The dielectric interaction between the CNT/BNNT layers induced by the vibrational motions can be related to the electron transfer. So that, another possible mechanism of electron transfer is the atomic motion triggering flexoelectric effects[46,47].

Overall, we have demonstrated that hot electrons induce the fast emission of radial phonon enhanced by an electron transfer channel at the interface between the interface of CNTs and BNNTs in a 1D vdW heterostructure. The physical picture is supported by the complementary use of ultrafast broadband transient spectroscopy, UED, and first-principles calculations. UED measurements reveal that the electron transfer between the CNT to BNNT layers observed by transient optical spectroscopy, accelerates the radial expansion of the 1D vdW heterostructure. The combined use of different techniques helps to decipher the fundamental physical processes resulting from interfacial interactions in vdW heterostructures and their photoinduced dynamics. These results suggest that due to the peculiar nature of this class of materials, being a vdW-heterostructure, carriers can be injected into the large bandgap BNNT thanks to the mixing of electronic states at interfaces. These insights are needed in order to push their limits for potential applications in ultrafast electronic, optoelectrical, and photothermal fields. Furthermore, these results pave the way for a better understanding of photoinduced electronic and structural dynamics in nanometric material, crucially needed for future photoactive devices and manipulation of photocarriers.

## Methods
### Synthesis of CNTs and CNT-BNNT heterostructures
CNT forests were synthesized using a chemical vapor deposition (CVD) method using Black Magic II (Aixtron Ltd.). The iron thin film (1.5 nm) was deposited on the substrate ($Al_2O_3/SiO_2/Si$) by electron beam deposition using VTR-350M/ERH (ULVAC KIKO Inc.). The substrate was annealed at a temperature of 350 °C in an atmosphere of hydrogen gas, which transform the iron thin film to iron nanoparticles on the substrate. After the particle formation, acetylene gas (100 sccm) with hydrogen gas (1000 sccm) and nitrogen gas (1000 sccm) was introduced into the synthesis chamber. At the same time of gas injection, the substrate was annealed to approximately 560 °C to grow the high dense CNT forest. The CNT sheet was drawn from the CNT forest by the dry-spinning method[33].

The CNT-BNNT heterostructures were prepared by synthesized boron nitride on a CNT sheet based on the powder CVD method. A quartz tube with two heating regions was used for a CVD setup for BNNT synthesis. The CNT sheet was placed at the center of the quartz tube and heated at 1050 °C using an electric furnace. Borane–ammonia ($NH_3BH_3$) (20 g) was placed in the quartz tube 28.5 cm away from the CNT sheet as a raw material for BNNTs. $NH_3BH_3$ was heated at 110 °C using a ribbon heater. $Ar/H_2$ gas flowed in the tube during the synthesis. BNNTs were synthesized for 3–6 h. After BNNT growth, the CNT-BNNT heterostructure sheet was immediately cooled to room temperature[34]. To make the same condition with the CNT sheet and CNT-BNNT heterostructure, the CNT sheet was also prepared by heated in the CVD setup for 3 h without putting $NH_3BH_3$. BNNTs without CNT core were also prepared to place the CNT-BNNT heterostructure (BNNT synthesis for 3 h) in the center of the quartz tube and heated at 650 °C for 1 h under atmospheric conditions, where the CNT core was removed. Layer numbers and diameters of CNT sheet and BNNT-grown CNT sheets are shown in Supplementary Figs. 3 and 4, observed with a transmission electron microscope. The CNTs and CNT-BNNT sheets are mounted on a sample holder made by

a $Si_3N_4/Si$ substrate (see Supplementary Fig. 1) for following ultrafast broadband transient absorption spectroscopy and UED measurements.

### Ultrafast broadband transient absorption spectroscopy
Broadband transient absorption spectroscopy (Fig. 6a) is performed on CNTs and CNT-BNNT heterostructures at room temperature using an ultrafast transient absorption spectrometer (HARPIA-TA, LIGHT CONVERSION) with a Yb doped potassium gadolinium tungstate (Yb:KGW) laser (PHAROS, LIGHT CONVERSION). The fundamental optical pulse from the Yb:KGW laser at a central wavelength of 1030 nm operating at a repetition rate of 10 kHz is split into two lines, pump and probe. The wavelength of the pump pulse is set to 400 nm with an optical parametric amplifier (OPA) (ORPHEUS, LIGHT CONVERSION). The fluence of the pump pulse is set to 3 mJ $cm^{-2}$. Broadband supercontinuum is generated by a yttrium aluminum garnet (YAG) or sapphire crystals depending on the probed spectral range, 1200–1600 nm and 360–1200 nm, respectively. The broadband probe pulse is spectrally resolved using optical grating and a complementary metal oxide semiconductor (CMOS) detection. Transient absorption spectra are recorded by changing the delay between pump and probe pulses. The experimental temporal resolution (full width of half maximum (FWHM)) is estimated at around 170 fs. The light polarization axes of pump and probe pulses are colinear with the longitudinal direction of the CNTs and BNNTs.

Ultrafast time-resolved absorption spectroscopy for deep-UV probing is performed on CNTs and CNT-BNNT heterostructures at room temperature using a laboratory-built setup (Fig. 6b) based on Ti:sapphire laser with a central wavelength of 800 nm at a repetition rate of 1 kHz. The light pulse is separated into pump and probe lines. The 400 nm pump pulse is produced by second harmonic generation in a BBO crystal (β-$BaB_2O_4$). The frequency of the pump is set to 500 Hz using an optical chopper. The fluence of the pump pulse is set to 3 mJ $cm^{-2}$. For the probe, an optical pulse with a deep-UV light with a wavelength of 230 nm (5.4 eV) is generated by an OPA. After passing through the optical delay stage, the probe pulse is transmitted by the sample and detected with a photodiode. The transient absorption changes are recorded by changing delays and reading the transient signal with lock-in detection. The experimental time resolution (FWHM) is approximately 200 fs. Both pump and probe polarization are colinear with the long axis of the CNTs and BNNTs. The efficient dynamics is observed with parallel polarization inducing efficient excitation of carriers in CNT.

### Ultrafast time-resolved electron diffraction measurements
UED measurements are performed on CNTs and CNT-BNNT heterostructures at room temperature using a laboratory-built setup (Fig. 7). Optical pulse with a central wavelength of 800 nm, duration of 100 fs, pulse energy of 2.0 mJ at a repetition rate of 1 kHz are produced by a regenerative amplifier (Spitfire XP, Spectra-Physics) and separated into pump and probe pulses using a beam splitter. The pump pulse (400 nm) with an estimated pulse duration of 100 fs is produced by second harmonic generation into a BBO crystal and then focused on the sample to induce photoexcitation at fluences of 1–4 mJ $cm^{-2}$. The wavelength for the probe line was set to 267 nm using BBO and calcite crystals. The probe pulse is focused onto the Au photocathode to generate bunches of electrons through the photoelectric effect. The kinetic energy of the electron is set to 75 keV. An electron pulse contained approximately $3 \times 10^4$ electrons. The diameter of the electron pulse was approximately 100 μm, and the pulse duration was <1 ps. Electron bunches diffract with the sample and are measured by a charge-coupled device (CCD) camera coupled to a magnetic lens. The repetition rate of the system (optical and electron pulses) is 1 kHz, and the exposure time for one image is set to 1 s. For each delay, ten images

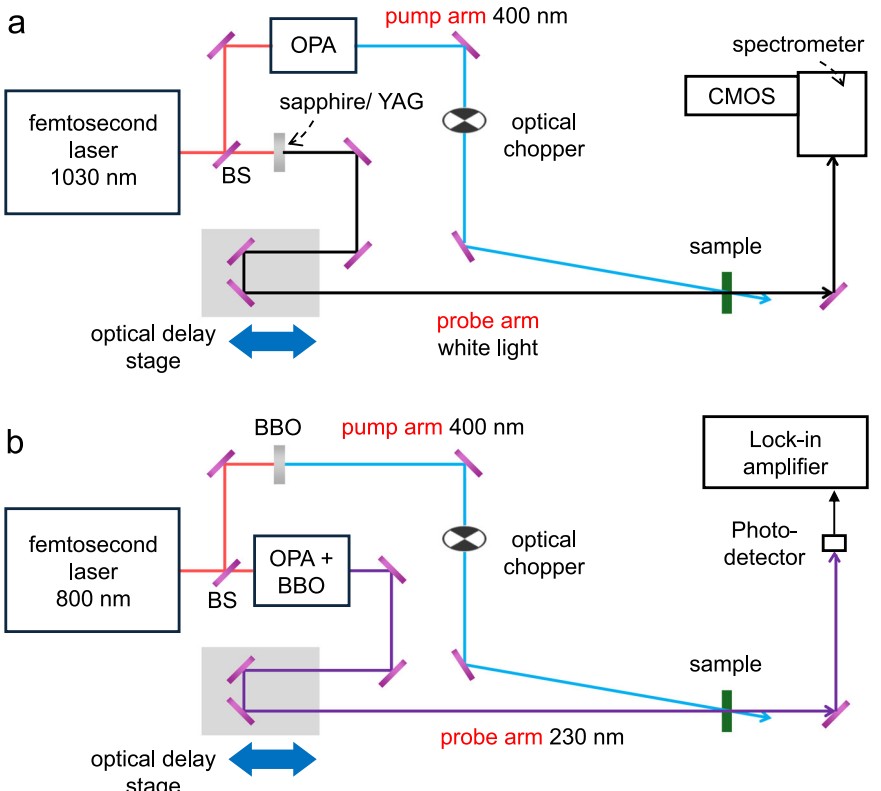

**Fig. 6 | Experimental setups for transient absorption spectroscopy.** The wavelengths of the probe light are in the near-UV to near-IR region (**a**) and deep-UV region (**b**). BS, BBO, YAG, and CMOS are the beam splitter, β-barium borate crystal, yttrium aluminum garnet crystal, and complementary metal oxide semiconductor camera, respectively.

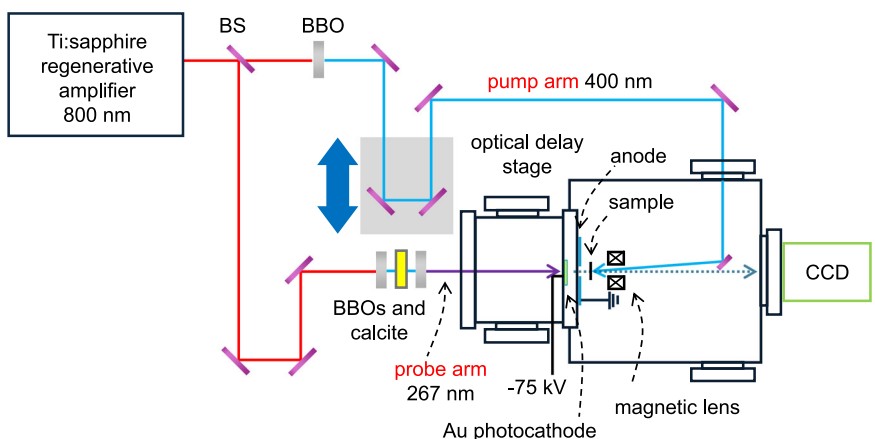

**Fig. 7 | Schematic illustration of ultrafast time-resolved electron diffractometer.** BS, BBO, and CCD are the beam splitter, β-barium borate crystal, and charge-coupled device camera, respectively.

are averaged translating into $3 \times 10^8$ electrons per delays. The samples of CNTs and CNT-BNNT heterostructures used for the UED are the same as for ultrafast broadband transient absorption spectroscopy.

**Calculations for electronic structure of multi-walled CNTs**
We used Atomic Simulation Environment (ASE)[48] for modeling multi-walled CNTs. We considered (10,0)@(19,0)@(28,0)CNT (228 atoms) with a diameter of 21.9 Å (Supplementary Table 1). The interlayer distance between the inner and outer nanotubes is almost the same as the graphite interlayer distance (~3.4 Å). We performed geometry optimization for multi-walled CNTs by using Quantum ESPRESSO (QE) code[49].

The exchange-correlation energy was treated within the generalized gradient approximation of Perdew, Burke, and Ernzerhof[50]. The electron-ion interaction was treated by using ultrasoft pseudopotential in the pslibrary.1.0.0[51]. The cutoff energies of the wavefunction and charge density were set to be 55 Ry and 550 Ry, respectively. The van der Waals correction within the DFT-D3 level[52] was included to describe the interlayer distance between inner and outer nanotubes. We assumed multi-walled CNTs in a tetragonal cell, where the lattice parameters of $a = b = 40$ A were fixed, and the length of $c$ was optimized. A $1 \times 1 \times 16$ k grid and a smearing parameter of 0.15 Ry[53] were assumed in the self-consistent field calculations. The electronic band

structure was calculated at 50 k points along the ΓX line. To plot the density-of-states (DOS), we used a Gaussian broadening of 0.2 eV. The calculated band structure and DOS for multi-walled CNTs are shown in Supplementary Fig. 11.

### First principles calculations for dynamics of electron transfer

The electronic states of model structures were calculated using the projected-augmented wave (PAW)[54,55] method within the framework of the density functional theory (DFT). We included the 2 s and 2p states as the valence electrons. The generalized gradient approximation formulated by Perdew, Burke, and Ernzerhof (GGA-PBE)[50] was used for the exchange-correlation potential. The cut-off energies of the plane-wave were 30 and 250 Ry for the electronic pseudo-wave functions and pseudo-charge density, respectively. The energy functional is minimized with respect to the Kohn–Sham (KS) orbitals using an iterative method[56,57]. We described excited electronic states as linear combinations of electron-hole pairs within Casida's linear-response time-dependent density functional theory (LR-TDDFT)[58], using the ground-state KS orbitals as a basis set. In LR-TDDFT, electronic excitation energies are calculated from the poles of an electron-hole pair response function. This procedure amounts to solving an eigenvalue problem, with a matrix size of $N_O N_u \times N_O N_u$ when using the GGA, where $N_O$ and $N_u$ are the numbers of occupied and unoccupied KS orbitals, respectively, used to represent excited states. To calculate excited states, we include all occupied states whose energy differences from LUMO are within 4 eV and all unoccupied states whose energy differences from HUMO are within 4 eV. Here, many-body effects are introduced by coupling matrix elements consisting of the random-phase-approximation and exchange-correlation terms. Molecular dynamics (MD) simulations of the canonical ensemble were performed using the Nosè–Hoover thermostat technique[59,60]. The equations of motion were integrated numerically using an explicit reversible integrator[61] with a time step of 50 a.u. (~1.2 fs). All MD simulations were performed at room temperature. The model structure of h-BN and graphene, single-walled CNT and single-walled BNNT, and double-walled CNT and single-walled BNNT are shown in Supplementary Figs. 19–21. To investigate the effects of photoexcitation on the atomic motion of the graphene/h-BN heterostructure, we performed nonadiabatic quantum molecular dynamics (NAQMD)[62] based on TDDFT calculations. Information for numerical details of MD calculations are shown in Supplementary Note 1. The initial and final configurations of MD calculations are in separate Supplementary Data files (Supplementary Datasets 1 and 2). All atomic configurations used in Fig. 4 are also shown as separate Supplementary Data files (Supplementary Datasets 3–6).

### Reporting summary

Further information on research design is available in the Nature Portfolio Reporting Summary linked to this article.

## Data availability

Source data generated in this study have been deposited in Figshare with the identifier "https://doi.org/10.6084/m9.figshare.25728837". All relevant experimental and computational data within the manuscript are available from the corresponding author upon request.

## Code availability

The computational codes within the manuscript are available from the corresponding author upon request. The first principles calculations in this study were performed based on the TDDFT method. The source code for the calculations is available as QXMD code available in https://usccacs.github.io/QXMD/index.html. (Permanent link to the version of code/repository for the TDDFT method can also be found in https://github.com/ElsevierSoftwareX/SOFTX_2019_49)[63].

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

## Acknowledgements

This work was supported by JSPS Kakenhi Grants-in-Aid [Nos. JP18H05208 (S.K.), JP22KK0225 (M.H.) and JP23H01101 (M.H.)] and the Japan Science Technology Agent (JST) FOREST Program [No. JPMJFR211V (M.H.)]. R.B. acknowledges Agence Nationale de la Recherche (ANR) for funding undergrant, ANR-21-CE30-0011-01 CRITI-CLAS. This work was also supported by Dynacom [International Research Laboratory between the CNRS, the University of Tokyo and the University of Rennes (R.B.)].

## Author contributions

M.H. conceived the idea and designed the study by discussing with H.S. Y.S., H.S., R.S., Y.I., M.K., Y.T., T.T., and Y.H. carried out the synthesis and static measurements. Y.S., T.G., Y.I., N.G., G.P., R.B., and M.H. performed the transient optical absorption experiments. Y.S., H.S., R.S., M.K., W.Y., and M.H. performed the ultrafast time-resolved electron diffraction measurements. S.Ohmura, S.Ono, and K.T. performed the first principles calculations. Y.S., T.G., R.S., G. N., R.B., and M.H. performed the data analysis. Y.S., H.S., S.Ohmura, S.K., K.T., R.B., and M.H. interpreted the data. Y.S., H.S., S.Ohmura, R.B., and M.H. wrote the paper. All authors discussed the results and edited the paper.

## Competing interests

The authors declare no competing interests.
