## [Peer Review File · Nature Communications]

Photoinduced dynamics during electronic transfer from narrow to wide bandgap layers in one-dimensional heterostructured materialsReviewer #1 (Remarks to the Author):

In the manuscript "Photoinduced dynamics generated by unusual electron transfer channels in a one-dimensional van der Waals heterostructure", the authors investigated the the electron transfer process in one-dimensional CNT-BNNT van der Waals heterostructures using femtosecond broadband optical spectroscopy, ultrafast time-resolved electron diffraction, and first-principles theoretical calculations. The results obtained are interesting and should be published. However, before publication, I hope you can make some revisions and deeper analysis. Specific comments are as follows:

(i) On page 16, the authors mention that "Due to the high calculation cost for the case of 1D heterostructure, we used the approximation of a two-dimensional heterostructure with the periodic boundary condition." What is the reason why the authors use 2D heterostructure instead of 1D case?

(ii) In fact, the curvature effect of 1D structures is the most important factor on the charge transfer process. The authors should be clarified.

(ii) What is the effect of flexoelectricity on the electron transfer process in 1D van der Waals heterostructures?

(iii) Figure 5 should not be set in the conclusion section. Also, the conclusion needs to be further condensed.

(iv) In order to enlarge the views of flexoelectric effect and one-dimensional van der Waals heterostructure, the following reference may be helpful (Phys. Rev. B, 2023, 108, 045416; Phys. Chem. Chem. Phys., 2021, 23, 20574).

Reviewer #2 (Remarks to the Author):

In this submission the authors present a set of results using ultrafast electron diffraction and ultrafast transient absorption spectroscopy to probe electronic processes in multi-walled carbon nanotube / boron nitride nanotube heterostructures. This topic is particularly interesting as the 1D van der Waals materials area seems to be undergoing a bit of a renaissance at present, building on some of the exciting things being done in 2D vdW materials. The authors present a set of results from complex experiments and should be complimented on attempting to combine results from ultrafast electron diffraction and transient absorption, which is a novel and exciting idea, along with theory from time-dependent density functional theory. However there are a large number of problems with the interpretation of the experimental and theoretical results as presented at present that makes me question the validity of the conclusions drawn. Most of these could be addressed by a major re-write of the paper, combined with a few extra experiments.

One issue I have with the results presented in the manuscript is that a variety of basic materials characterisation that seems to be missing. For example it is common to show the radial breathing modes of carbon nanotubes via Raman spectroscopy to establish their diameter, but this has not been mentioned here. That would have been an obvious thing to do to check the CNT diameter, and also to look at if the presence of the outer BNNTs changes the radial breathing modes of the CNTs.

From the information stated it is not clear if the nanotubes themselves are very high quality. The TEM images (low resolution) show a lot of variation in the side-walls of the material labelled as BNNTs. Was that all BN? Or organic contaminants? There are no element-specific TEM (e.g. EELs) maps shown.

There is information missing that would help understand their results as well. The stated diameter of their nanotubes is "5nm" in the text – but it is not clear if this is the diameter of each individual nanotube, the diameter of the outer-most CNT in a multi-walled CNT, or the diameter of a bundle of smaller (e.g. 1nm) nanotubes. The diameter is very important for understanding the optical properties and electronic properties of CNTs.

My main issue is that at present I don't think the authors have presented a clear physical picture of what happens after light absorption in their nanotubes. Regard of the specifics of the CNT diameter, the pictures of the bandstructure in Figure 2d and Figure 5 is incorrect. With 3.1eV

incident energy and wider CNTs there is no chance that absorption transfers an electron between the linear Dirac bands. A more accurate picture of the bandstructure of the CNTs should be drawn, as their bandstructure is very well known. For instance at 3.1eV the pump will be photoexciting electrons into a multitude of curved bands with minima well above the Dirac point (see e.g. any of the many books and articles on the bandstructure of CNTs). There are a number of points further on through the manuscript where the physical description of the processes that are occurring is inaccurate or in need of being further developed (see comments below about lines 148, 152, 221).

The UV excitation at 400nm of the CNTs is in the tail of the well-known pi-plasmon peak of CNTs at around 270nm, which corresponds to collective oscillations of pi-electrons. This mode is broad and is on top of the impurity band often found in graphitic carbon systems. BNNTs are also well known to have strong collective pi-electron plasmons (see e.g. Margulis et al., Physical Review B 78, 035415). In light of these well established facts, I think it strange that the paper does not consider or discuss the physics of pi-plasmons, which can cause inter-tube Coulomb coupling (Margulis's paper).

In terms of physical interpretation and discussion of their transient absorption results, the authors should also refer to work on intertube excitons in single-walled carbon nanotubes with BNNTs and MoS₂ NTs (Adv. Funct. Mater. 2104969), where photoexciting the CNTs with infrared light is reported to form intertube excitons in the MoS₂, very similar to the transient absorption results here, which could be interpreted as a near-UV pulse absorbed by the CNT creating an intertube exciton across the CNT/BNNT interface.

To me, the authors seem to observe via ultrafast electron diffraction that after photoexcitation of the CNTs both the CNTs and BNNTs have expanded. This is entirely expected (thermal expansion). Did the authors perform a temperature-dependent electron diffraction experiment, for example heating the sample by ~100 degrees and re-measuring the electron diffraction pattern? This should be straightforward to do as CNTs and BNNTs are both robust to high temperatures.

Finally, the authors advance some first principles theory based on a 2D model of graphene coupled to hBN. However, graphene is not an appropriate system to use to model this experiment as (1) the 2D analogue of the material should be graphite, or multi-layer graphene, as the experiment was performed with multi-layered CNTs, not with single-walled CNTs; (2) graphene in 2D does not have the pronounced excitonic features seen in 1D carbon nanotubes because the Coulomb force's effect in 2D is quite different.

Lines 52-54: motivating why electron transfer processes in vdW heterostructures are interesting. References 8-13 are cited, but it is not explained to the reader what the unique characteristics are?

Line 110-111: it would be better to say the transient absorption is negative (i.e. the absorption decreases after photoexcitation) rather than "the transient absorption decreases" as stated.

Line 112 – should read "generation of free carriers and excitons" inside the CNTs, as the primary photoproduct in semiconducting (and also metallic) CNTs is most often an exciton.

Line 115 – all pump-probe signals are formally a non-linear optical process (e.g. multidimensional coherent spectroscopy tells us standard pump-probe signals are often a $\chi^{(3)}$ process). I suggest changing "exclude a non-linear optical process as being responsible" to "exclude multi-photon absorption from contributing".

Line 148: the authors claim that "some of the photoexcited electrons transfer to the conduction band of BNNTs through a peculiar electron transfer channel provided by the 1D character of the material, and the others relax back near to the Fermi-level". There are two issues with this sentence: firstly the nanotubes are most likely not particularly 1D: as the absorption spectra seem to show there are no clear van Hove / excitonic resonances, which are normally clear if the material has 1D electronic behaviour. If the CNTs are multi-walled or bundled, the electrons are

free to move both along the CNT's axis and in the radial direction. Secondly, there is nothing "peculiar" about the transfer process: there are a number of physical mechanisms that explain charge or energy transfer between nearby nanoparticles / nearby molecules (e.g. Forster resonant energy transfer, quantum tunnelling).

Line 152: Electrons transferred to the BNNT cannot recombine by radiative recombination within the BNNT: there should be no holes in the BNNT - as grown, it should be an intrinsic semiconductor. The bandgap of BNNT is so large that it is probably quite hard to dope BNNT, unless there are some very shallow states. There is no evidence that the BNNT produced are p-doped.

Line 158 and 160: Caption label says "transient absorption spectrum" for panel (c), but this is not a spectrum, it is the dynamics ("time trace" as used elsewhere).

Figure 3: "Intensity" misspelt in Figure 3e.

Line 221-222: "the relaxation process occurs in ~ 1 ps. During this process, phonons can be generated on the BNNTs side to relax the transferred free carriers...". This 1ps time refers to the lifetime of the transient change in absorption at 230nm, shown in Figure 2c, which presumably relates to the timescale for the electron to be removed from the conduction band of the BNNT, after which the interband absorption has returned to normal. What is meant by these lines was not clear: did the authors mean that during the ~ 1 ps, while there are free charges in the conduction band of the BNNT, these hot electrons can cool by emitting phonons within the BNNT? If so, I agree, but it is not clear from what is written. If by "relax" the authors meant that electrons in the BNNTs return to the unoccupied electronic state in the valence band of the CNT by phonon emission, then this is not clear how this would work.

Lines 293-294: The main conclusion that "a peculiar electron transfer channel induces radial phonon emission at the interface between the interface of CNTs and BNNTs in a one-dimensional vdW heterostructure" is not supported by the experimental results or the interpretation presented. The physics of the "peculiar" transfer is not described. It is not shown that radial phonons play a particularly significant role: electrons in nanotubes can also relax via tangential phonon emission. Transferring heat from the electronic system to the lattice would also be expected to heat the lattice, causing an expansion of the nanotubes.

Lines 308-310: It also doesn't make sense that there is anything particularly unique about the 1D vdW heterostructure. If the TDDFT models and interpretation are correct, which were on a periodic 2D system, then these effects can be seen in 2D (the theory presented) as well as 1D (claimed to be presented in the experiment, although this is also not clear given the multi-walled nature of the CNTs and BNNTs). The interfacial mixing of electronic states and/or Coulomb coupling between layers is expected to be strong, but it is not a phenomenon unique to 1D vdW materials.

Reviewer #3 (Remarks to the Author):

In this manuscript, the authors proposed a near-ultraviolet photoexcitation induced electron transfer channel in the CNT-BNNT heterostructures. The transient absorption spectrum probed at 230 nm showed an unusual dynamics feature which is not observed in pure CNTs. The author attribute it to the charge transfer inside the heterostructures within ~ 1 ps. Moreover, the ultrafast electron diffraction shows that the interlayer expansion is accelerated with the appearance of CNTs-BNNTs interface and the increased number of BNNT layer, which was attributed to the phonons generation on the BNNTs side and charge delocalization caused by the electron transfer. Furthermore, the proposed scenario was consolidated by TD-DFT.

This work expanded the research of electron transfer process in vdW heterostructures from 2D down to 1D world. The experimental measurements show good agreement with the calculation results. In this sense, this paper is very timely and well aligned with the current efforts in the field of ultrafast science in low-dimensional van der Waals heterostructures. However, there are still some comments should be addressed.

1. For saving computing time, the theoretical calculations were carried out based on the 2D approximation with the periodic boundary condition, and provided consistent results with experiments. Is that possible for similar experimental phenomena being reproduced in 2D graphene-BN heterostructures?
2. As mentioned in the main text, it's the anisotropic energy/charge transfer makes the 1D heterostructures interesting and differ from 2D case. Does the transfer channel uncovered in this work show such kind of feature?
3. The transient absorption around 365 nm in Fig. 2a goes down before time zero and goes up at positive time delays, which appears to be different from the rest wave band. So the related discussion is need in the text about this obvious feature.
4. The nanotubes were well aligned along one direction in this work. Previous studies have shown that the polarization of pump can affect the electroacoustic coupling efficiency of CNTs (Nanoscale, 2022,14, 10477-10482). The authors did not give any information about the polarization state of the light used in this paper. It is possible that alteration of polarization direction could affect the experimental data.
5. The evolution of interlayer distance in different samples were provided in the normalized manner in Fig. 3f. As a suggestion, the original data could be included in the Supplementary material. I believe this will have some significance for other researchers to refer to the article.
6. If the three sets of points in Fig. 3f were fitted with equation 2, the solid lines should start to rise at the same point (time zero).

Reply to Reviewer #1

In the manuscript “Photoinduced dynamics generated by unusual electron transfer channels in a one-dimensional van der Waals heterostructure”, the authors investigated the the electron transfer process in one-dimensional CNT-BNNT van der Waals heterostructures using femtosecond broadband optical spectroscopy, ultrafast time-resolved electron diffraction, and first-principles theoretical calculations. The results obtained are interesting and should be published. However, before publication, I hope you can make some revisions and deeper analysis. Specific comments are as follows:

We are grateful that the reviewer highly evaluates our work. We also thank the reviewer for fruitful suggestions to improve the quality of the manuscript. Answers to comments are attached below.

- (i) On page 16, the authors mention that “Due to the high calculation cost for the case of 1D heterostructure, we used the approximation of a two-dimensional heterostructure with the periodic boundary condition.” What is the reason why the authors use 2D heterostructure instead of 1D case?
- (ii) In fact, the curvature effect of 1D structures is the most important factor on the charge transfer process. The authors should be clarified.

We thank the reviewer for these comments. The comments (i) and (ii) are essential and asked by other reviewers. We have performed theoretical calculations on a one-dimensional (1D) van der Waals (vdW) interface between a carbon nanotube (CNT) and a boron nitride nanotube (BNNT) with various diameters. We found that electrons cannot transfer from CNT to BNNT with a small diameter (less than 1 nm) at their interface. However, a similar effect as one observed in a two-dimensional (2D) vdW interface between graphene and boron nitride is reproduced at the CNT and BNNT interface above the diameter of 1.6 nm, with the occurrence of the interlayer electron transfer. The outer diameter of CNTs used in our experiment is ~5 nm; therefore, the situation is even closer to the case of a 2D vdW interface. We have revised this additional calculation as revised Fig. 4 and Supplementary Figs. 24–28.

- (iii) What is the effect of flexoelectricity on the electron transfer process in 1D van der Waals heterostructures?

We sincerely thank the reviewer for this thoughtful comment. The vibrational motions are induced in the CNT and BNNT layers upon photoexcitation. The local vibrational motions

occur quite fast, which is estimated to be less than 500 fs by theoretical calculations. Some parts of the distance between the CNTs and BNNTs decrease, and the others increase. The dielectronic interaction between the layers can be related to the electron transfer. Thus, the flexoelectricity effect, so that the local bending of materials induces dielectric effects, may be one of the fundamental mechanisms for understanding the electron transfer at the vdW heterostructures between CNTs and BNNTs. We have referred to the possibility of flexoelectricity in the main text.

(iv) Figure 5 should not be set in the conclusion section. Also, the conclusion needs to be further condensed.

We have rearranged the figures in the results and discussion parts. We have also further condensed the conclusion.

(v) In order to enlarge the views of flexoelectric effect and one-dimensional van der Waals heterostructure, the following reference may be helpful (Phys. Rev. B, 2023, 108, 045416; Phys. Chem. Chem. Phys., 2021, 23, 20574).

We have added the relevant references regarding flexoelectricity in the main text (references 45 and 46).

Reply to Reviewer #2

In this submission the authors present a set of results using ultrafast electron diffraction and ultrafast transient absorption spectroscopy to probe electronic processes in multi-walled carbon nanotube / boron nitride nanotube heterostructures. This topic is particularly interesting as the 1D van der Waals materials area seems to be undergoing a bit of a renaissance at present, building on some of the exciting things being done in 2D vdW materials. The authors present a set of results from complex experiments and should be complimented on attempting to combine results from ultrafast electron diffraction and transient absorption, which is a novel and exciting idea, along with theory from time-dependent density functional theory. However there are a large number of problems with the interpretation of the experimental and theoretical results as presented at present that makes me question the validity of the conclusions drawn. Most of these could be addressed by a major rewrite of the paper, combined with a few extra experiments.

We thank the reviewer for this positive statement. We also thank the reviewer for careful review and experimental suggestions that improve the quality of the manuscript. The answers to the comments are attached below.

One issue I have with the results presented in the manuscript is that a variety of basic materials characterisation that seems to be missing. For example it is common to show the radial breathing modes of carbon nanotubes via Raman spectroscopy to establish their diameter, but this has not been mentioned here. That would have been an obvious thing to do to check the CNT diameter, and also to look at if the presence of the outer BNNTs changes the radial breathing modes of the CNTs.

We thank the reviewer for the suggestion. We have performed the Raman spectroscopy on CNT and CNT-BNNT samples (revised Supplementary Figs. 8 and 9). Peaks around the G-band suggest the creation of BNNT. However, the radial breathing mode region shows a broad spectrum because the sample very likely contains CNTs and BNNTs with different diameters and chirality. The average diameter of our CNTs is approximately 3–5 nm, and that of BNNTs is even greater. According to Kataura's plot, very broad spectra are expected when the sample contains large-diameter CNTs. Instead of using spectroscopic tools, we have deduced an average diameter of CNTs and BNNTs from TEM images, as shown in revised Supplementary Fig. 4.

From the information stated it is not clear if the nanotubes themselves are very high quality. The TEM images (low resolution) show a lot of variation in the side-walls of the material labelled as

BNNTs. Was that all BN? Or organic contaminants? There are no element-specific TEM (e.g. EELs) maps shown.

We thank the reviewer for the suggestion. We have now performed TEM-EELs measurements on the CNT-BNNT samples and identified that the inner part of the sample consists of carbon and the outer part of the sample consists of all boron and nitrogen. We have added these experiments to the supplementary materials (revised Supplementary Figs. 5–7).

There is information missing that would help understand their results as well. The stated diameter of their nanotubes is “5nm” in the text – but it is not clear if this is the diameter of each individual nanotube, the diameter of the outer-most CNT in a multi-walled CNT, or the diameter of a bundle of smaller (e.g. 1nm) nanotubes. The diameter is very important for understanding the optical properties and electronic properties of CNTs.

We agree with the reviewer’s suggestion. We have counted the diameter of CNTs as a histogram from TEM images, as shown in supplementary Fig. 4. The inner and outer diameters of the CNTs are approximately 3 and 5 nm, respectively. Thus, the diameter at the interface of the CNTs is approximately 5 nm, and that of counter BNNTs (innermost layer of BNNTs) is estimated to be approximately 6 nm (approximately $5 + 0.34 + 0.34$ nm).

My main issue is that at present I don’t think the authors have presented a clear physical picture of what happens after light absorption in their nanotubes. Regard of the specifics of the CNT diameter, the pictures of the bandstructure in Figure 2d and Figure 5 is incorrect. With 3.1eV incident energy and wider CNTs there is no chance that absorption transfers an electron between the linear Dirac bands. A more accurate picture of the bandstructure of the CNTs should be drawn, as their bandstructure is very well known.

We do agree that we use a very simplified (maybe oversimplified) picture of the electronic band structure of CNT. In fact, optical experiments and further calculations have revealed the impact of the multi-gap-character (like band semiconductors) in the out-of-equilibrium electronic dynamics in pure CNT. We now used a more elaborate and realistic picture to describe the electronic band structure.

For instance at 3.1eV the pump will be photoexciting electrons into a multitude of curved bands with minima well above the Dirac point (see e.g. any of the many books and articles on the bandstructure of CNTs). There are a number of points further on through the manuscript where the physical description of the processes that are occurring is inaccurate or in need of being further

developed (see comments below about lines 148, 152, 221).

We do agree that the previous representation was oversimplified, and we now use a more accurate one. We revised the model presentation and rewrote the manuscript accordingly.

The UV excitation at 400nm of the CNTs is in the tail of the well-known pi-plasmon peak of CNTs at around 270nm, which corresponds to collective oscillations of pi-electrons. This mode is broad and is on top of the impurity band often found in graphitic carbon systems. BNNTs are also well known to have strong collective pi-electron plasmons (see e.g. Margulis et al., Physical Review B 78, 035415). In light of these well established facts, I think it strange that the paper does not consider or discuss the physics of pi-plasmons, which can cause inter-tube Coulomb coupling (Margulis's paper).

Thank you very much for the suggestion. For the CNT, a tail of the π -plasmonic peak (or impurity band) may be excited with 400 nm. The Coulombic interaction, which may be induced by the interlayer electron transfer related to the π -plasmonic excitation, might be a possible mechanism of interlayer expansion observed by ultrafast time-resolved electron diffraction measurements. Regarding BNNT, since we could not observe any structural changes from ultrafast time-resolved electron diffraction measurements upon near-UV photoexcited BNNTs, we assume that the π -plasmonic peak of BNNT could not be efficiently excited. We have referred to the π -plasmonic interaction and inter-tube Coulombic coupling effects in the conclusion.

In terms of physical interpretation and discussion of their transient absorption results, the authors should also refer to work on intertube excitons in single-walled carbon nanotubes with BNNTs and MoS₂ NTs (Adv. Funct. Mater. 2104969), where photoexciting the CNTs with infrared light is reported to form intertube excitons in the MoS₂, very similar to the transient absorption results here, which could be interpreted as a near-UV pulse absorbed by the CNT creating an intertube exciton across the CNT/BNNT interface.

We appreciate the suggestions from the reviewer. The intertube excitation across the CNT/BNNT interface is another possible mechanism to describe the physical phenomena into play. We have added this discussion to the main text.

To me, the authors seem to observe via ultrafast electron diffraction that after photoexcitation of the CNTs both the CNTs and BNNTs have expanded. This is entirely expected (thermal expansion). Did the authors perform a temperature-dependent electron diffraction experiment, for example heating the sample by ~100 degrees and re-measuring the electron diffraction pattern?

This should be straightforward to do as CNTs and BNNTs are both robust to high temperatures. Indeed, the structural dynamics agree with the simple picture of energy dissipation into the lattice subsystem. What remains unclear is whether the phononic subsystem clearly follows a Bose-Einstein distribution at an early time, allowing for an estimation of temperature. Also, the main point of the diffraction experiment is to show that structural dynamics are going faster in the case of 1D vdW compared to the pure CNT case, very likely through the presence of an electron transfer channel or ‘intertube’ exciton. Thermal transfer generally occurs in a few to several picoseconds. If the thermal transfer occurs at the vdW interface, the interlayer expansion of pure CNTs should be faster than 1D vdW heterostructures since the light is absorbed only by the CNT side. We have also measured the electron diffraction pattern at 74°C compared to those at 24°C (see below). The reciprocal space of interlayer distance of 74°C and 24°C are both 0.282 \AA^{-1} , which are identical to each other, indicating a very small thermal expansion in this range (within the sensitivity of our apparatus). However, the observed photoinduced structural changes consist of an expansion from 0.282 \AA^{-1} to 0.279 \AA^{-1} . This result suggests that the photoinduced effects of the change of the interlayer distance are much more significant than the one expected from a simple temperature jump induced solely by heat transfer at the interface.

Finally, the authors advance some first principles theory based on a 2D model of graphene coupled to hBN. However, graphene is not an appropriate system to use to model this experiment as (1) the 2D analogue of the material should be graphite, or multi-layer graphene, as the experiment was performed with multi-layered CNTs, not with single-walled CNTs; (2) graphene in 2D does

not have the pronounced excitonic features seen in 1D carbon nanotubes because the Coulomb force's effect in 2D is quite different.

We thank the reviewer for the comment. We have performed additional first-principles calculations on 1D vdW systems, *i.e.*, a single-walled CNT covered with single-walled BNNT (revised Fig. 4 and Supplementary Figs. 24 and 26) and double-walled CNT covered with a single-walled BNNT (Supplementary Figs. 25, 27, and 28). The results can answer the question. We found that the electron can transfer from a single-walled CNT to a single-walled BNNT in a relatively large diameter (more than 1.6 nm) at their interface, similar to the results of the 2D system (graphene and h-BN). The diameter of CNTs we used for the experiment was ~ 5 nm. Here, to understand the stacking effect of CNTs (since we use few-walled CNTs covered with few- or multi-walled BNNTs), we also calculated the double-walled CNTs covered with single-walled BNNTs. The results are similar to the case of single-walled CNT covered with single-walled BNNT. The effect of stacking CNTs is much smaller than that of the interfacial diameter of CNTs, which is consistent with a previous report [*Phys. Rev. B* **85**, 085411 (2012)]. Furthermore, we also have referred to the possibility of the excitonic dynamics between the CNTs and BNNTs.

Lines 52-54: motivating why electron transfer processes in vdW heterostructures are interesting. References 8-13 are cited, but it is not explained to the reader what the unique characteristics are?

We thank the reviewer for the comment. Previously, the static functions of stacked low-dimensional materials were mainly discussed; however, the present study derives a new function from the photoexcited non-equilibrium state. This is the main characteristic of this study. We have rewritten the relevant parts of the introduction.

Line 110-111: it would be better to say the transient absorption is negative (*i.e.* the absorption decreases after photoexcitation) rather than “the transient absorption decreases” as stated.

We changed the statement accordingly to improve its clarity.

Line 112 – should read “generation of free carriers and excitons” inside the CNTs, as the primary photoproduct in semiconducting (and also metallic) CNTs is most often an exciton.

We do agree with the reviewer. We revised it accordingly.

Line 115 – all pump-probe signals are formally a non-linear optical process (e.g. multidimensional coherent spectroscopy tells us standard pump-probe signals are often a $\chi^{(3)}$ process). I suggest changing “exclude a non-linear optical process as being responsible” to “exclude multi-photon absorption from contributing”.

We thank the reviewer and have revised the passage accordingly to improve clarity. As the reviewer figures out, the main point was to remove the possibility of multi-photon absorption for being responsible for the observed effects.

Line 148: the authors claim that “some of the photoexcited electrons transfer to the conduction band of BNNTs through a peculiar electron transfer channel provided by the 1D character of the material, and the others relax back near to the Fermi-level”. There are two issues with this sentence: firstly the nanotubes are most likely not particularly 1D: as the absorption spectra seem to show there are no clear van Hove / excitonic resonances, which are normally clear if the material has 1D electronic behaviour. If the CNTs are multi-walled or bundled, the electrons are free to move both along the CNT’s axis and in the radial direction. Secondly, there is nothing “peculiar” about the transfer process: there are a number of physical mechanisms that explain charge or energy transfer between nearby nanoparticles / nearby molecules (e.g. Forster resonant energy transfer, quantum tunnelling).

As the reviewer points out in the first part, our sample consists of bundled few-walled carbon nanotubes, the electronic structure of which is not purely a 1D system. The calculated density of the state of a three-walled carbon nanotube is shown in revised Supplementary Fig. 11, where a clear van Hove singularity does not exist. Thus, our sample behaves not like 1D pure CNTs but like band semiconductors. We revised these sentences more accurately. Secondly, the charge transfer itself is not unusual since it may occur by the interlayer exciton or the local contact induced by the layer vibration; however, injecting the electron into the band structure of BNNT with 400 nm light without two-photon absorption is worth mentioning in this context.

Line 152: Electrons transferred to the BNNT cannot recombine by radiative recombination within the BNNT: there should be no holes in the BNNT - as grown, it should be an intrinsic semiconductor. The bandgap of BNNT is so large that it is probably quite hard to dope BNNT, unless there are some very shallow states. There is no evidence that the BNNT produced are p-doped.

We agree with the reviewer that there is no evidence that the BNNT is a p-typed semiconductor. Because we can inject electrons or excitons into the band of a BNNT with 400 nm light pump, excluding a two-photon absorption process, there should be a local contact between the electronic bands of the CNT and BNNT. We have revised it accordingly.

Line 158 and 160: Caption label says “transient absorption spectrum” for panel (c), but this is not

a spectrum, it is the dynamics (“time trace” as used elsewhere).

We thank the reviewer for catching this mistake and change accordingly.

Figure 3: “Intensity” misspelt in Figure 3e.

We thank the reviewer for catching this mistake and change accordingly.

Line 221-222: “the relaxation process occurs in ~ 1 ps. During this process, phonons can be generated on the BNNTs side to relax the transferred free carriers...”. This 1ps time refers to the lifetime of the transient change in absorption at 230nm, shown in Figure 2c, which presumably relates to the timescale for the electron to be removed from the conduction band of the BNNT, after which the interband absorption has returned to normal. What is meant by these lines was not clear: did the authors mean that during the ~ 1 ps, while there are free charges in the conduction band of the BNNT, these hot electrons can cool by emitting phonons within the BNNT? If so, I agree, but it is not clear from what is written. If by “relax” the authors meant that electrons in the BNNTs return to the unoccupied electronic state in the valence band of the CNT by phonon emission, then this is not clear how this would work.

We appreciate the comment from the reviewer. We have added the following: “While there are free charges in the conduction band of the BNNT, these hot electrons can cool by emitting phonons within the BNNT within 1 ps. This 1 ps timescale refers to the dynamics of the abrupt transient optical change monitors at 230 nm.” Indeed, this is the statement we want to stress because it may also be a reasonable explanation for the system expanding faster in this case due to strong electron-phonon interactions in BNNT inducing faster structural changes compared to the pure CNT case.

Lines 293-294: The main conclusion that “a peculiar electron transfer channel induces radial phonon emission at the interface between the interface of CNTs and BNNTs in a one-dimensional vdW heterostructure” is not supported by the experimental results or the interpretation presented. The physics of the “peculiar” transfer is not described. It is not shown that radial phonons play a particularly significant role: electrons in nanotubes can also relax via tangential phonon emission. Transferring heat from the electronic system to the lattice would also be expected to heat the lattice, causing an expansion of the nanotubes.

Indeed, free charges will equilibrate via phonon-emission through classical electron-phonon coupling in CNT and BNNT parts of the 1D vdW heterostructure, generating both radial and tangential phonons. They are somehow a byproduct of the electronic relaxation of the material, and our apparatus is more sensitive to structural dynamics modifying interlayer distances. Anyhow, it is very likely that the phononic subsystem does

not reach a thermal distribution at an early time scale as non-thermal phonon distribution is mentioned in numerous materials up to several tens of picoseconds after femtosecond optical excitations. A key point will be whether the observed structural changes are as fast as the observed ones if no electrons can be transferred between the two sub-parts of these vdW heterostructures. In addition, the observed magnitude of the structural changes seems larger than a simple “temperature” jump of a few tens of degrees. Suppose the simple heat vibrational transfer occurs at the interface between the 1D vdW heterostructures; photoexcitation of CNTs and subsequent heat transfer to BNNT shows slower structural dynamics. However, the observed phenomenon was the opposite, suggesting fast energy transfer (electron or excitonic transfer) at the interface of 1D vdW heterostructures. We have added this explanation in the conclusion.

Lines 308-310: It also doesn't make sense that there is anything particularly unique about the 1D vdW heterostructure. If the TDDFT models and interpretation are correct, which were on a periodic 2D system, then these effects can be seen in 2D (the theory presented) as well as 1D (claimed to be presented in the experiment, although this is also not clear given the multi-walled nature of the CNTs and BNNTs). The interfacial mixing of electronic states and/or Coulomb coupling between layers is expected to be strong, but it is not a phenomenon unique to 1D vdW materials.

We agree with the reviewer that our observation works not only particularly for 1D vdW systems but also for 2D vdW systems. The main point of using 1D vdW systems for ultrafast time-resolved electron diffraction measurements is to access both intra and interlayer dynamics in the normal geometry while having an almost flat Ewald sphere. It turns out to be relevant as most of the observed dynamics impact the interlayer distances. As mentioned above, the multi-walled nature has been considered in the revised manuscript. We first revised the introduction to emphasize the advantages of using the 1D system for the measurements and revised the conclusion part, which is not specific to the 1D case.

Reply to Reviewer #3

In this manuscript, the authors proposed a near-ultraviolet photoexcitation induced electron transfer channel in the CNT-BNNT heterostructures. The transient absorption spectrum probed at 230 nm showed an unusual dynamics feature which is not observed in pure CNTs. The author attribute it to the charge transfer inside the heterostructures within ~ 1 ps. Moreover, the ultrafast electron diffraction shows that the interlayer expansion is accelerated with the appearance of CNTs-BNNTs interface and the increased number of BNNT layer, which was attributed to the phonons generation on the BNNTs side and charge delocalization caused by the electron transfer. Furthermore, the proposed scenario was consolidated by TD-DFT.

This work expanded the research of electron transfer process in vdW heterostructures from 2D down to 1D world. The experimental measurements show good agreement with the calculation results. In this sense, this paper is very timely and well aligned with the current efforts in the field of ultrafast science in low-dimensional van der Waals heterostructures. However, there are still some comments should be addressed.

We thank the reviewer for the positive evaluation of our work. We also thank him/her for fruitful suggestions to improve the quality of the manuscript. The answer to the comment is attached below.

1. For saving computing time, the theoretical calculations were carried out based on the 2D approximation with the periodic boundary condition, and provided consistent results with experiments. Is that possible for similar experimental phenomena being reproduced in 2D graphene-BN heterostructures?

We would like to answer “yes” to the question. We would expect a similar effect in 2D graphene-BN heterostructures. There is a geometrical (regarding experimental apparatus) challenge to observe the interlayer dynamics of 2D materials using ultrafast time-resolved electron diffraction. There is another challenge caused by the scattering strength of graphene and h-BN that consist of light elements. However, we believe similar effects will be reproduced in 2D materials to improve the experimental setup for future works.

We have also performed theoretical calculations on a 1D interface between a CNT and a BNNT with various diameters to make the answer more clearly. We found that the electron cannot transfer at a wavelength of 400 nm from a CNT to a BNNT in smaller diameters (less than 1 nm) at their interface. Above the diameter of 1.6 nm, however, a similar effect as the 2D interface between graphene and boron nitride is reproduced at a

CNT and BNNT interface, where an interlayer electron transfer occurs. This may answer this question. The outer diameter of CNTs used in our experiment is ~5 nm; therefore, the situation is even closer to the case of the 2D interface. We have added the additional calculations in the revised Fig. 4 and Supplementary Figs. 24–28.

2. As mentioned in the main text, it's the anisotropic energy/charge transfer makes the 1D heterostructures interesting and differ from 2D case. Does the transfer channel uncovered in this work show such kind of feature?

Thank you very much for pointing this out. In general, the anisotropic energy transfer of 1D heterostructures is interesting and differs from the 2D case. However, we believe the energy transfer channels discovered in this work would be observed in the 2D case as mentioned above. From the perspective of measurements, it is quite challenging to observe the interlayer dynamics of a 2D system using ultrafast time-resolved electron diffraction as the Ewald sphere is almost flat, inducing a much lower sensitivity of out-of-plane/interlayer dynamics. The 1D structure provides easier access to interlayer dynamics via ultrafast time-resolved electron diffraction measurements. Observed physical views in the 1D system are complementarily measured with transient optical spectroscopy and supported by first-principles calculations in the present study. We have emphasized this statement in the manuscript to make it clear.

3. The transient absorption around 365 nm in Fig. 2a goes down before time zero and goes up at positive time delays, which appears to be different from the rest wave band. So the related discussion is need in the text about this obvious feature.

We acknowledge the reviewer for this remark. As the reviewer points out, there is a positive residual signal after the negative peaks in the UV region (above the energy of the pump pulse), which differs from the transient absorption in the visible to infrared region. This feature is closer to the transient absorption at the wavelength of 230 nm, which shows a positive peak and subsequent negative residual signal. Thus, the band renormalization induced by electron transfer or interlayer exciton may be responsible for the small positive residual signal after the negative peaks in the UV region (~365 nm). We have added the discussion in the manuscript.

4. The nanotubes were well aligned along one direction in this work. Previous studies have shown that the polarization of pump can affect the electroacoustic coupling efficiency of CNTs (Nanoscale, 2022,14, 10477-10482). The authors did not give any information about the polarization state of the light used in this paper. It is possible that alteration of polarization

direction could affect the experimental data.

This information was provided in SI and is also added to the main text for clarity. “Both pump and probe polarization are colinear with the long axis of the CNT and BNNT. The efficient dynamics is observed with parallel polarization inducing efficient excitation of carriers in CNT.” Actually, no clear signal was observed in optical spectroscopy with the perpendicular configuration. Ultrafast time-resolved electron diffraction measurements and optical spectroscopy used the same polarization configurations to keep experimental parameters as similar as possible. We have also performed ultrafast time-resolved electron diffraction measurements on a CNT sample using near-UV photoexcitation with a polarization axis perpendicular to the long axis of the CNT; however, we could not observe the signal like the near-UV photoexcitation with a polarization axis parallel to the long axis of the CNT. This is simply because the CNTs hardly absorb the light of which polarization is perpendicular to the long axis of the CNTs.

5. The evolution of interlayer distance in different samples were provided in the normalized manner in Fig. 3f. As a suggestion, the original data could be included in the Supplementary material. I believe this will have some significance for other researchers to refer to the article.

Thank you very much for the comment. The original data is in the revised Supplementary Fig. 19.

6. If the three sets of points in Fig. 3f were fitted with equation 2, the solid lines should start to rise at the same point (time zero).

Thank you very much for catching this feature. We revised the figure accordingly.

Reviewer #1 (Remarks to the Author):

The related issues have been solved, and I recommend accepting this paper.

Reviewer #2 (Remarks to the Author):

In this substantially revised resubmission the authors have responded to all the requests made in the previous round. The paper does now look more robust. In particular I like the improved discussion of the optoelectronic properties of the nanotubes. I fully support publication. One minor correction "dielectronic" around line 369 should be "dielectric".

Reviewer #3 (Remarks to the Author):

In the revised version of the manuscript, the authors have significantly improved the quality of the discussion and satisfactorily addressed my comments. I recommend the publication of the manuscript after the minor comments below are addressed.

1. The comparison of transient absorption between pure CNT and CNT-BNNTS heterojunction presented in Fig. S17 is a good indicator of the existence of charge transfer. It could be merged into Figure 2c in the main text.
2. For clarity, the diffraction spots and rings in Fig. 3a-c should be indexed.
3. Line 132: 'within <1 ps' → 'within 1 ps'.
4. Line 427: "Supplementary Fig. 16" is not the setup revised sketch.

Reply to Reviewer #1

The related issues have been solved, and I recommend accepting this paper.

We are grateful that the reviewer's recommendation for publications of our work.

Reply to Reviewer #2

In this substantially revised resubmission the authors have responded to all the requests made in the previous round. The paper does now look more robust. In particular I like the improved discussion of the optoelectronic properties of the nanotubes. I fully support publication. One minor correction "dielectronic" around line 369 should be "dielectric".

We thank the reviewer for his/her support. We revised "dielectronic" to "dielectric".

Reply to Reviewer #3

In the revised version of the manuscript, the authors have significantly improved the quality of the discussion and satisfactorily addressed my comments. I recommend the publication of the manuscript after the minor comments below are addressed.

We appreciate the reviewer's recommendation for publication of our work.

1. The comparison of transient absorption between pure CNT and CNT-BNNTS heterojunction presented in Fig. S17 is a good indicator of the existence of charge transfer. It could be merged into Figure 2c in the main text.

We moved the Supplementary Fig. 17 to the main text (Fig. 2c), according to the reviewer's advice.

2. For clarity, the diffraction spots and rings in Fig. 3a-c should be indexed.

We indexed them in the caption accordingly.

3. Line 132: 'within <1 ps' → 'within 1 ps'.

We revised "within <1 ps" to "within 1 ps".

4. Line 427: "Supplementary Fig. 16" is not the setup revised sketch.

We corrected the numbers of Supplementary Figures.